# Chitosan/Pomegranate Seed Oil Emulgel Composition as a New Strategy for Dermal Delivery of Hydrocortisone

**DOI:** 10.3390/ijms25073765

**Published:** 2024-03-28

**Authors:** Zofia Helena Bagińska, Magdalena Paczkowska-Walendowska, Anna Basa, Michał Rachalewski, Karolina Lendzion, Judyta Cielecka-Piontek, Emilia Szymańska

**Affiliations:** 1Student Scientific Group, Department of Pharmaceutical Technology, Medical University of Bialystok, Mickiewicza 2c, 15-222 Białystok, Poland; baginska.zosia@gmail.com; 2Department of Pharmacognosy and Biomaterials, Poznan University of Medical Sciences, Rokietnicka 3 Str., 60-806 Poznań, Poland; mpaczkowska@ump.edu.pl (M.P.-W.); jpiontek@ump.edu.pl (J.C.-P.); 3Faculty of Chemistry, University of Bialystok, Ciołkowskiego 1K, 15-245 Białystok, Poland; abasa@uwb.edu.pl; 4Dr Irena Eris, Centre for Science and Research, Armii Krajowej 12, 05-500 Piaseczno, Poland; michal.rachalewski@drirenaeris.com (M.R.); karolina.lendzion@drirenaeris.com (K.L.); 5Department of Pharmaceutical Technology, Medical University of Bialystok, Mickiewicza 2c, 15-222 Białystok, Poland

**Keywords:** emulgel, chitosan, pomegranate seed oil, hydrocortisone, penetration enhancer, anti-inflammatory activity, multifunctional drug carrier

## Abstract

Multifunctional delivery systems capable of modulating drug release and exerting adjunctive pharmacological activity have attracted particular attention. Chitosan (CS) and pomegranate seed oil (PO) appear to be attractive bioactive components framing the strategy of complex therapy and multifunctional drug carriers. This research is aimed at evaluating the potential of CS in combination with PO in studies on topical emulgels containing hydrocortisone as a model anti-inflammatory agent. Its particular goal was to distinguish alterations in anti-inflammatory action followed with drug dissolution or penetrative behavior between the designed formulations that differ in CS/PO weight ratio. All formulations favored hydrocortisone release with up to a two-fold increase in the drug dissolution rate within first 5 h as compared to conventional topical preparations. The clear effect of CS/PO on the emulgel biological performance was observed, and CS was found to be prerequisite for the modulation of hydrocortisone absorption and accumulation. In turn, a greater amount of PO played the predominant role in the inhibition of hyaluronidase activity and enhanced the anti-inflammatory effect of preparation E-3. Emulgels showed a negligible reduction in mouse fibroblasts’ L929 cell viability, confirming their non-irritancy with skin cells. Overall, the designed formulation with a CS/PO ratio of 6:4 appeared to be the most promising topical carrier for the effective treatment of inflammatory skin diseases among the tested subjects.

## 1. Introduction

Topical corticosteroids are commonly used as the first line of therapy in a range of inflammatory topical diseases, including contact or atopic dermatitis and psoriasis. The mode of action is broad and involves anti-inflammatory, anti-mitotic, and immunosuppressive effects. Topical corticosteroids markedly decrease the number of infiltrating cells, namely mast cells, macrophages, eosinophils, and T-lymphocytes in the inflamed tissue [1,2]. At present, approximately 30 compounds differing in potency and clinical efficacy and classified by strength and the risk of adverse effects are available on the pharmaceutical market [2,3,4]. Hydrocortisone (HTZ) is a low potent steroid agent and is sparingly soluble in aqueous media (250 μg/mL) with log P 1.61 [5]. HTZ, the drug of choice for the treatment of mild skin irritation and rashes, is considered a safe, well-tolerated therapeutic option, including for children. HTZ is currently available in conventional formulations, including ointments, creams, and lotions [6].

In recent years, much attention has been devoted to smart, multifunctional drug delivery systems in which bioactive compounds formulating the drug carrier simultaneously exhibit adjunctive biological activity (e.g., antioxidant, anti-inflammatory, anti-microbial) and therefore may act as an active part of the therapy [7,8]. In addition, by controlling the release pattern and/or enhancing the permeability of the active agent across biological membranes, the drug carrier is considered to improve its bioavailability [9,10]. Chitosan (CS) is a naturally-derived, biodegradable polycationic polysaccharide with *N*-acetylglucosamine and glucosamine-functional groups. Due to their anti-inflammatory, antimicrobial activity and ability to promote skin healing, CS and its derivatives have been extensively studied in the pharmaceutical technology of topical formulations [11]. When applied topically, CS may act as penetration enhancer, facilitating drug diffusion by opening tight junctions or by interacting with extra-cellular matrix components [12,13]. Therefore, CS appears to be a promising bioactive component in keeping with the strategy of multifunctional drug carriers and complex therapy. Pomegranate seed oil (PO) is a cold-pressed oil obtained from pomegranate tree seeds. With regard to feasible nourishing and emollient properties, it is considered an attractive cosmetic ingredient that can be used to improve skin elasticity, maintain proper skin hydration, and stimulate cell regeneration. PO contains a variety of active ingredients including unique punicic acid (representing conjugated fatty acids), phenolic acids, flavonoids, phytosterols, or gamma-tocopherol, which are responsible for anti-inflammatory and skin-repairing properties [14]. At present, the antiproliferative potential of PO is being particularly explored for the prevention and treatment of breast, prostate, colon, and skin cancer [15,16].

Emulgel is a semi-solid formulation comprised of an oil in water or a water in oil emulsion combined with the gel base. This combination is characterized by its greaseless, transparent appearance and favorable application properties including its moisturizing, emollient and soothing effects [17,18]. In recent years, emulgels have been particularly tested for their ability to modulate drug release and permeability across the skin barrier [19,20,21]. Currently, emulgels are being tested as delivery platforms for anti-inflammatory drugs [22], antifungal agents [23], and antibiotics [21]. Several pharmaceutical products are commercially available in the form of emulgel, including Clinagel^®^ (clindamycin phosphate), Miconaz-H^®^ (miconazole nitrate), Pernox^®^ (benzoyl peroxide), and Voltaren Emulgel^®^ (diclofenac sodium) [5].

Some research papers have recently drawn attention to the use of CS as a gelling agent in emulgel compositions for the management of skin dermatitis, wound healing, and tissue regeneration [24,25,26]. To our best knowledge, there are a limited number of studies devoted to the application of CS and PO or pomegranate extract in food or cosmetic technology [27,28] and no research data focused on the use of CS and PO in the pharmaceutical technology of drug delivery systems. 

Therefore, this study aimed to assess the potential of CS in combination with PO in the development of studies on topical emulgel containing HTZ as a model anti-inflammatory agent. Designed formulations differing in CS to PO weight ratios were evaluated for physicochemical and textural properties in terms of their applicability and usefulness as skin delivery platforms. A precise effort was made to determine the optimal formulation by examining the effect of CS to PO weight ratio on HTZ dissolution and penetration behavior as well as anti-inflammatory activity measured by in vitro hyaluronidase inhibition assay. The assessment of irritation potential by agarose overlay assay in a mouse fibroblast NCTC 929 cell line was also carried out.

## 2. Results and Discussion

### 2.1. Emulgels Characteristics

The current approach of pharmaceutical technology focuses on developing multifunctional carriers with the ability to modulate drug release and facilitate the pharmacological efficacy of active agents. The favorable biological properties of CS and PO led us to investigate their potentials as components of topical emulgels containing HTZ as a model anti-inflammatory agent. In the present studies, the four formulations of E-1–E-4 differing in their weight ratios of CS and PO were developed using the homogenization technique (Table 1).

The prepared emulgels were non-transparent preparations with smooth, homogeneous appearances and gently nutty scents. The presence of HTZ gave the preparations an off-white color except for formulation E-3, which was slightly yellowish due to a greater amount of PO present within its composition. The examined pH values ranged from 5.7 to 5.9, which lie within the physiological pH range of the skin and should not cause any risk of skin irritation (Table 2). Solubility studies showed that about 25% of the drug embedded in the emulgel was in a dissolved state (Table 2). There was no sign of phase breakdown in any of the designed formulations except for sample E-4, which tended to separate over the first 24 h. The weight ratio of the CS base to the PO (4:6) in formulation E-4 resulted in the spontaneous irreversible loss of its homogenous structure with the simultaneous precipitation of drug particles. Upon centrifugation, the separation of the oil and the hydrophilic phase was noticed for samples E-3 and E-4, revealing their limited physical stability even in the presence of two emulsifier agents. In contrast, emulgels E-1 and E-2 were found uniform with no presence of biliquid layers under stress conditions. Due to impaired stability and spontaneous phase separation shortly after preparation, emulgel E-4 was excluded from further examinations.

The representative transmission electron microscopy (TEM) micrographs for samples E-1 and E-2 showed the presence of spherical oil phase droplets with relatively narrow size distributions (Figure 1). The emulgels presented submicron oil droplet sizes with almost no particles larger than 5 μm. Basically, the size of oil droplets was found irrespective of the applied weight ratios of CS to PO, as no real differences in the oil phase diameters were noticed between the tested E-1 and E-2. Notably, no drug particles larger than 1 μm were observed, and no sign of their aggregation was noticed within the tested emulgel samples, confirming that the HTZ was carefully dispersed and homogeneously incorporated within formulations.

### 2.2. Textural Behavior

To provide deeper insight into the internal structures of emulgels, the textural studies of the formulations were assessed and the firmness, cohesiveness, and consistency measurement data are displayed in Figure 2.

In semi-solid formulations, the firmness parameter describes the sample hardness, cohesiveness reflects the sample’s potential recovery after mechanical stress, and consistency defines the sample’s spreading ability on the skin surface. Dermatological drug preparations with poor textural behavior may be considered undesirable for topical delivery as they may wash out rapidly from the application site. The tested HTZ-loaded formulations responded differently to compression stress, and the ratio of CS to PO impacted the textural behavior of the emulgels. As shown in Figure 2a,c, by decreasing the ratio between the CS and PO (from 8:2 to 6:4), gradual increases in hardness and consistency values were observed in preparations E-1 and E-2. This observation contrasts with previous findings in which a decrease in the amount of CS in the formulation was responsible for a drop in its rheological behavior [29,30]. Emulgel E-2 exhibited the highest values of mechanical properties among those tested. This was most likely the result of stronger ionic crosslinking between positively charged CS chains and carboxylic groups of unsaturated fatty acids presented in PO. This interaction imparted the CS chains’ configuration and reduced their flexibility, leading to a more packed structure formation [31]. A degree of physical modification and interaction strength is affected, among other characteristics, by the ratio between the polymer and the crosslinking agent. In the present work, the CS/PO weight ratio 6:4 was found to form a sufficiently dense network of ionic bridges which substantially increased the mechanical properties of emulgel E-2. Both preparations E-1 and E-2 were relatively firm, presenting comparable values of cohesiveness (Figure 2b). This compact structure reflects stronger interactions between the carrier ingredients and should support drug retention at the application site. In turn, emulgel E-3 with a CS/PO weight ratio of 2:8 proved to be the weakest in respect to textural properties. A dominant presence of the liquid oil phase in formulation E-3 resulted in approximately two- and three-fold drops in hardness, cohesiveness, and consistency when compared to formulations E-1 and E-2. Importantly, profound differences were noticed between the designed emulgels and the control, which was thecommercially available preparation with HTZ. Basically, the control studies revealed the greater values of mechanical properties for the reference material except for the consistency parameter (Figure 2c), which was found to be comparable with formulation E-2. This data is in line with our assumption and confirmed the lighter textures of the emulgels when compared to the conventional cream formulation.

### 2.3. In Vitro Dissolution Profile

Next, the influence of the CS/PO weight ratio in emulgel compositions on the HTZ dissolution rate was studied. A comparison between the HTZ release pattern from the designed emulgels is presented in Figure 3.

In terms of the topical delivery of an anti-inflammatory agent, it is important to ensure a relatively fast and steady drug release profile to initiate drug action on the skin. A clear impact of CS/PO weight ratio on the HTZ release behavior was noticed in this study. Within the first stage of the test (Figure 3a), the amount of the drug released from the designed formulations ranged from 8% to 20% of the total drug dose, and formulation E-2 (with CS/PO weight ratio 6:4) demonstrated the fastest initial release rate among the tested samples. Interestingly, the dissolved HTZ fraction observed in the acceptor medium was found to be irrespective of the CS ratio in the carrier composition, and emulgels E-1 and E-3 with reversed weight ratios of CS/PO (8:2 vs. 2:8) displayed comparable release patterns within the first 5 h of study. According to pharmacopeia guidelines [32], dissolution studies were continued until at least 80% of the drug appeared in the acceptor medium (t80%). As shown in Figure 3b, the dissolution lasted up to 96 h and faster releases of HTZ were observed for CS/PO emulgels than for the commercially available product. Emulgel E-2 showed the highest release of HTZ with more than 80% of drug released within 60 h of the test. Slower drug dissolution was achieved for emulgel E-1, which reached t80% after about 80 h. The observed delay in the HTZ release rate from preparation E-1 (with a CS/PO ratio of 8:2) resulted from a greater amount of CS, which was responsible for the formation of a thicker, gel-like barrier for diffusing drug particles in an acceptor environment with pH 5.5 (Figure 3b). This is in accordance with previous findings where the presence of CS in the drug carrier was associated with a prolonged release pattern and an extended dissolution rate of the active substances [33,34]. Interestingly, control studies revealed profoundly lower onsets, with less than 15% of drug release within the 96 h period (Figure 3b). This most likely resulted from the more compact structure and greater textural properties observed in the reference preparation (Figure 2). Surprisingly, formulation E-3, which had the lowest textural behavior, displayed retarded dissolution, and 80% of its HTZ was released after 96 h. A slower pattern of dissolution observed in the studies with formulation E-3 was most likely due to the presence of the hydrophobic oil matrix, which hindered drug diffusion by limiting the entrance of the acceptor fluid.

The obtained findings indicated the complex nature of HTZ release from CS/PO emulgels, which depends both on the CS to PO weight ratio and the mechanical properties of the tested formulations. Importantly, the form of an emulgel appeared to favor HTZ release as compared to conventional topical preparations, with formulation E-2 showing a greater drug release rate at earlier time points, which in turn may result in a faster onset of anti-inflammatory action.

### 2.4. Penetration and Retention Studies

In vitro penetration studies are important tools for predicting in vivo topical absorption [35]. In the present study, we compared the penetration behavior of HTZ from emulgels E-1, E-2, and E-3 across a skin biomimetic membrane by using a Skin PAMPA^TM^ kit. The applied model solely assessed the passive drug diffusion. According to [34], compounds with *P_app_* < 1 × 10^−6^ cm/s are classified as having low permeability and those with *P_app_* > 1 × 10^−6^ cm/s as having high permeability. To evaluate the influence of single formulation components on HTZ permeability, additional control studies with pure CS gel base with HTZ (Control 1), PO with HTZ (Control 2), and pure drug in a buffer solution (Control 3) were carried out. Table 3 displays the cumulative amount of HTZ permeated over time and expressed as an apparent permeation coefficient.

The penetration rate of HTZ was found to be time-dependent, and relatively low HTZ diffusion across the biomimetic membrane was observed for all samples at an early experimental stage. After 4 h, the permeability coefficients of HTZ from the emulgels were below 1 × 10^−6^ cm/s for sample E-3 (with a CS/PO weight ratio 2:8), together with Control 2 displaying relatively the highest P_app_ values among the tested samples. In contrast, a greater amount of HTZ was found in the acceptor medium after a 24 h incubation, and P_app_ was above 1 × 10^−6^ cm/s for all emulgel samples. The highest drug penetration was observed for formulation E-1, followed by E-2 and E-3, which was almost 10 times higher than that observed for the control studies with the pure HTZ (Control 3). Some of the previous literature data indicate the greater permeability of HTZ from gels as compared to conventional topical formulations [35], which confirms the validity of the formulation applied in this study.

Both of the used ingredients, PO and CS, are penetration enhancers that promote drug diffusion across the skin membrane [13,36]. In our studies, the abilities of CS/PO emulgels to enhance the penetration rate of HTZ was clear, but only in the later time points. Interestingly, none of the control studies, including the pure CS with HTZ (Control 1) and the PO base with HTZ (Control 2), exceeded 1 × 10^−6^ cm/s for *P_app_* (Table 3), suggesting that only CS in combination with PO (and not the single ingredients) are necessary to improve HTZ transport across the skin epithelium. Notably, in the designed emulgel compostions, CS appeared to play a predominant role in modulating the HTZ absorption as formulation E-1 with a CS/PO ratio of 8:2 exhibited greater *P_app_* values among the tested samples.

After the active agent’s absorption into the skin layers, the epithelial cells act as a drug reservoir. Sufficient drug accumulation in the epithelium is important for maintaining its pharmacological activity, especially upon the clearance of the preparation from the skin’s surface. Therefore, drug retention studies were additionally performed using the StratM membrane to mimic the skin barrier. Figure 4 presents the cumulative amount of HTZ accumulated in the biomimetic membrane.

The degree of HTZ accumulation varied between 18 μg/cm^2^ and 30 μg/cm^2^, and the retention values attained for E-1 and E-2 were higher as compared to emulgel E-3 (*p* < 0.05 and *p* < 0.01, respectively). This observation shows that the greater amount of CS in a formulation is a prerequisite for improving tissue retention of the model anti-inflammatory agent. The obtained findings are in accordance with our previous studies on vaginal CS multiunit carriers in which higher amounts of CS in the formulations facilitated the tissue accumulation of the microbicide agent [37]. Notably, a higher HTZ tendency to accumulation was observed in the studies using E-2 (*p* < 0.01), in which a faster rate of drug dissolution was simultaneously observed (Figure 2). Interestingly, the developed emulgels, particularly E-1 and E-2, provided greater drug accumulations when compared to the pure drug control (C-3) (Figure 4). This observation supports data from the penetration studies (Table 2) and confirms the feasible potential of the CS/PO emulgel composition in modulating HTZ transport across skin. It should be noted that drug penetration and retention were not affected by the amount of the soluble HTZ fraction in formulations, as the solubility studies displayed comparable drug concentrations in dissolved states in all of the tested emulgels (Table 2). Overall, a certain effect of the CS to PO weight ratio on the penetration and accumulation behavior was noted, and formulations with greater amounts of CS (E-1 and E-2) were found to enhance drug permeability and retention in the biomimetic skin membranes.

### 2.5. Anti-Inflammatory Activity

Hyaluronic acid, a glycosaminoglycan in the epithelial tissue, is an important factor that decreases local inflammatory processes and influences skin tissue proliferation and differentiation [38]. Hyaluronic acid is hydrolyzed by a hyaluronidase enzyme present in the extracellular matrix during tissue remodeling. As upregulation of hyaluronidase activity occurs within chronic inflammatory processes, enzyme inhibitors are considered to play a beneficial role in the treatment of inflammatory disorders. In the present study, the focus was therefore on assessing the anti-inflammatory potentials of the designed emulgel compositions as measured by the degree of inhibition of the hyaluronidase enzyme activity. For this purpose, IC_50_ values referring to the emulgel concentrations needed to inhibit the enzyme activity by 50% were calculated (Table 4). Some research data reported that PO helped to regenerate skin tissue by inhibiting the activities of several enzymes including hyaluronidase [39], and the mode of CS anti-inflammatory action is related to polycationic behavior and changes in the secondary structure of this enzyme by protonated NH^3+^ groups in polymer chains.

As presented in Table 3, formulations E-2 and E-3 with CS/PO weight ratios of 6:4 and 2:8, respectively, showed greater anti-inflammatory activities, and their observed biological effects were found to be comparable between formulations. Unexpectedly, formulation E-1 with a CS/PO weight ratio of 8:2 was less effective as higher concentrations were needed to suppress hyaluronidase action. This observation is in contrast with the previously reported data, where the presence of CS in the drug carrier profoundly impacted hyaluronidase activity, and a clear correlation between the polymer concentration and the enzyme inhibition rate was noticed [40]. An additional control with pure HTZ in a concentration corresponding to that applied in designed formulations was studied concomitantly. The attained data (Table 4) demonstrated that the drug itself did not inhibit enzyme activity in the applied concentration and that only the emulgels’ ingredients were responsible for suppressing the hyaluronidase effect. The presented data showed that the anti-inflammatory effects of the emulgels varied in the CS to PO weight ratios, although PO was found to be prerequisite factor for inhibiting hyaluronidase activity and enhancing the anti-inflammatory response.

### 2.6. Agarose Overlay Assay 

The influences of emulgels E-1, E-2, and E-3, together with their drug-free counterparts on the viability of L929 cells, a commonly applied model for ISO cytotoxicity testing, are presented in Table 5. Based on the visual assessment after staining with MTT, the samples were graded and their qualifications above grade 2 (referring to the reduction of viability by more than 30%) were considered as cytotoxic effects.

Basically, the tested emulgels E-1–E-3 caused no damage to the L929 murine fibroblasts, and no lysis or reduction of cell growth was observed over a 24 h incubation. Some minor changes were observed only for emulgel E-1, in which slight signs of degeneration and the presence of loosely attached cells were noticed (Appendix A). In addition, the presence of HTZ in the preparations had no real effect on the cells’ metabolic activity as no differences in viability were observed when compared to the drug-free samples. The irritation potential was clear in the positive control studies, for which incubation with 3% SDS caused the visible destruction of cells and reductions in cell viability with a radius of lysis zone >2 cm around the filter disc (Table 5). Overall, the designed emulgels displayed comparable results to those attained for the negative control (PBS treated cells), confirming their non-irritancy and compatibility with skin cells.

## 3. Materials and Methods

### 3.1. Materials

Hydrocortisone was obtained from Fagron (Kraków, Poland). Highly purified medical grade chitosan (Chitoscience^®^) with an average molecular weight (232 kDa) measured by Agilent 1260 Infinity GPC/SEC with a refractive index detector (Agilent Technologies, Santa Clara, CA, USA) was purchased from Heppe Medical Chitosan GmbH (Halle, Germany). Deacetylation degree (79.2%) was determined by the titration technique according to [41]. Pomegranate seed oil was purchased from Chempol (Wrocław, Poland). Cetostearic acid was obtained from Paulika (Trąbki Wielkie, Poland). Soybean phosphatidylcholine (lecithin, Phospholipon 90) was obtained from Lipoid (Kőln, Germany). Bovine serum albumin (BSA), hyaluronic acid sodium salt, hyaluronidase from bovine testes, hexadecyltrimethylammonium bromide (CTAB), and phosphotungstic acid hydrate were purchased from Sigma Aldrich (Steinheim, Germany). Potassium dihydrogen phosphate, absolute alcohol, sodium laurylsulphate (SDS), and lactic acid were provided by Chempur (Piekary Śląskie, Poland). The reference topical preparation with 1% *w*/*w* hydrocortisone acetate (serial number 01AF0223), which was composed of water, propylene glycol, stearic acid, cetyl alcohol, paraffin, vaseline, sorbitan stearate, macrogol cetostearyl ether, and parabens M and P was obtained from Aflofarm (Pabianice, Poland). Water was prepared by a Milli-Q Reagent Water System (Millipore, Billerica, MA, USA). Acetonitrile and methanol (Merck, Darmstadt, Germany) were of HPLC-grade.

### 3.2. Preparation of Emulgels

Different formulations of emulgels comprised of 3% (*w*/*w*) chitosan in 1% (*w*/*w*) lactic acid as the aqueous phase and pomegranate oil in a weight ratio of 8:2 (E-1 o/w), 6:4 (E-2 o/w), 2:8 (E-3, w/o), or 4:6 (E-4 w/o) were prepared using the homogenization technique with the Heidolph Brinkmann Homogenizer Silent Crusher M (Heidolph Instruments GmbH, Schwabach, Germany). In brief, the hydrophilic phase was prepared by dissolving CS in lactic acid at 60 °C under continuous stirring (400 rpm). Next, lecithin was dispersed in the polymer solution (300 rpm, 50 °C), while the cetostearic acid was melted in pomegranate oil at 40 °C. HTZ (1%, *w*/*w*) mixed with glycerol (in a ratio 1:1, *w*/*w*) was then carefully suspended in the CS gel phase. The hydrophilic and lipid phases were gently heated to the same temperature and homogenized for 20 min at 15,000 rpm and then for additional 10 min at 10,000 rpm. All formulations were kept in closed containers at 2–8 °C prior measurements. The emulgels’ compositions are presented in Table 1.

The pH was measured in triplicate by a glass electrode from the pH-meter Orion 3 Star (ThermoScientific, Waltham, MA, USA). The soluble fraction of HTZ embedded in the emulgels was measured at 37  ±  0.5 °C in water. For this purpose, an excess of drug formulation was added to water, sonicated in an ultrasonic bath for 30 min, and incubated for 24 h in a water bath (37 °C and 150 rpm). Afterward, the samples were centrifuged to remove undissolved drug particles, filtered through a cellulose membrane filter (0.22 μm), and determined by the HPLC system (described in Section 3.3). All measurements were done in triplicate. The physical stability of the emulgels was investigated for macroscopic signs of phase separation by the centrifugation method (4000 rpm, for 30 min).

### 3.3. HTZ Content

The HTZ content was achieved with a reverse-phase, high-pressure liquid chromatography system using Agilent Technologies 1200 (Agilent, Waldbronn, Germany) according to European Pharmacopeia monograph [42] with modifications. For this purpose, the drug was extracted from the formulation with the freeze-thaw technique in a mixture of water, acetonitryl, and methanol (50:25:25, *v*/*v*). Each emulgel sample was vigorously shaken for 5 min, then incubated at 30 °C followed with a freezing procedure for 15 min at −20 °C. The extraction procedure was repeated three times. Separation was achieved on a Zorbax Eclipse C18-BDS, 5 μm, 4.6 × 150 mm column (Agilent, Santa Clara, CA, USA) at 254 nm by isocratic elution with water/methanol/acetonitrile (50:25:25, *v*/*v*) as the mobile phase. The flow rate was 1 mL/min and the column temperature was maintained at 25 °C. The calibration curves were linear in the range from 0.1 to 5 μg/mL (R_2_ = 0.9968) and from 5 to 100 μg/mL (R_2_ = 0.9955). The lowest limit of detection was 0.05 μg/mL. The intraday and interday precisions (%CV) for concentration 1 μg/mL were 0.6 and 1.1%, respectively.

### 3.4. TEM Analysis

The morphological study of the representative emulgel samples was examined using transmission electron microscopy (TEM, Tecnai-G2, 200 kV, HR-TEM, FEI, Eindhoven, The Netherlands) equipped with an electron source LaB_6_ and an ultrathin windowed energy dispersive X-Ray system. Prior to measurements, each formulation (10 μL) was placed on a carbon-film-coated 400 mesh copper grid, stained with 2% phosphotungstic acid, air-dried, and studied at 200 kV.

### 3.5. Textural Properties

The textural properties of the emulgels were measured by a Texture Analyser TA.XT.Plus (Stable MicroSystem, Godalming, UK) equipped with a backward extrusion measuring system. Each formulation (30.0 ± 0.1 g) was compressed with a disc (25 mm diameter) with a speed of 2 mm/sec into the sample (with a defined depth 10 mm) at ambient conditions. The textural characteristics were assessed as hardness (maximal force attained during the compression of emulgel), consistency (work required to deform the sample upon compression), and cohesiveness (force assessed upon the upward movement of the disc). During the measurements, the graph was plotted and the textural properties were calculated using Texture Exponent 32 software v. 6240. The results are shown as the averages of three independent experiments.

### 3.6. Dissolution Studies

The dissolution studies of the HTZ from the emulgels were carried out in a USP dissolution Apparatus II (Agilent 708 DS, Agilent Technologies, Cary, NC, USA) with enhancer cells (drug diffusion area of 3.80 cm^2^, Agilent Technologies, Cary, NC, USA) across a cellulose membrane (Cuprophan with molecular weight cutoff 10 kDa, Medicell, London, UK). About 1.0 g of each emulgel formulation (referring to 10 mg of HTZ) was placed in the enhancer cell, covered with cuprophan, and placed in the vessel with a phosphate buffer pH 5.5 (500 mL). The temperature was 32 ± 0.5 °C while the stirring rate was set at 75 rpm. At the predetermined time points, 2 mL of the acceptor medium was withdrawn, filtered through 0.2 μm cellulose acetate filters, and analysed using the HPLC method (as described in Section 3.3). Withdrawn medium samples were replaced with equal volumes of fresh buffer. A reference commercial cream with HTZ acetate registered for dermal delivery was applied in the control studies. All release experiments were conducted in triplicate.

### 3.7. Permeability Studies with Skin PAMPA Assay

The permeability of the emulgels was investigated by using the skin-parallel artificial membrane permeability assay Skin PAMPA™ (Pion Inc., Billerica, MA, USA). The model consists of a two-chamber PAMPA sandwich composed of two 96-well plates with a donor at the bottom and an acceptor at the top. The top plate contains the lipid-impregnated skin-mimetic membrane. Before use, the pre-coated Skin PAMPA™ sandwich plates were hydrated overnight by placing 200 μL of the hydration solution in each well (Hydration Solution, Pion Inc., Billerica, MA, USA). The emulgel samples were diluted with Prisma HT buffer, and an amount of 200 μL was placed in the donor compartment and incubated at 32 ± 1 °C. The experiments were performed at least three times using six replicates on each plate. The amounts of permeated HTZ after 4 and 24 h were determined using the HPLC method (Section 3.3). Additional samples, the CS gel base with HTZ, PO with HTZ, or pure HTZ (with drug concentration corresponding to its amount within tested emulgels), were applied as controls. The apparent permeability coefficient (*P_app_*) was calculated as follows:$$P_{app}=\frac{-ln(1-\frac{C_{A}}{C_{equilibrium}})}{S\times (\frac{1}{V_{D}}+\frac{1}{V_{A}})\times t}$$
where *V_D_*—donor volume, *V_A_*—acceptor volume, *C_equilibrium_*—equilibrium concentration, $C_{equilibrium}=\frac{C_{D}\;\times \;V_{D}\;+\;C_{A}\;\times \;V_{A}}{V_{D}\;+\;V_{A}},$
*C_D_*—donor concentration, *C_A_*—acceptor concentration, *S*—membrane area, and *t*—incubation time (measured in seconds).

### 3.8. Retention Studies

Retention studies were performed in in-line cell system equipped with thermostated diffusion chambers (SES GmbH Analysesysteme, Bechenheim, Germany) and Strat-M membrane (Merck Millipore Ltd., Darmstadt, Germany) imitating the human skin barrier properties [43]. Each emulgel, with a drug weight amount of 2 mg, was placed in the donor compartment on the top of the membrane. Next, the acceptor medium (0.9% saline solution with addition of 10% of ethanol to assure sink conditions (pH 7.0)) was recirculated beneath the membrane at a constant rate of 35 mL/h. After 24 h, each formulation was carefully removed, and the donor compartment was washed with a mobile phase until there was a complete removal of the drug from the top of the membrane. The biomimetic membrane was then transferred to the mobile phase, sonicated, and incubated in a thermostated water bath (40 °C) for 72 h. After being centrifuged (4000 rpm, 15 min) and filtered through nylon filters of 0.2 μm, the extract was examined for drug accumulation. Additional control studies with pure CS gel base with HTZ, PO with HTZ, or pure drug suspended in phosphate buffer were accomplished concomitantly. All studies were performed in triplicate.

### 3.9. Anti-Hyaluronidase Activity

Hyaluronidase inhibitions were determined by using the turbidimetric method according to [44] with reagent mixtures of 25 μL of hyaluronidase enzyme (30 U/mL in acetate buffer pH 7.0), 25 μL of acetate buffer (50 mM, pH 7.0, with 77 mM NaCl, and 1 mg/mL of BSA), 15 μL of acetate buffer (pH 4.5), and 10 μL of emulgels. All reagent mixtures were incubated at 37 ± 0.5 °C for 10 min. Next, a hyaluronic acid solution (0.3 mg/mL in acetate buffer pH 4.5) was added, and the mixtures were incubated at 37 ± 0.5 °C for 45 min. To precipitate the undigested enzyme, 200 μL of 2.5% CTAB in 2% NaOH was added and left at ambient conditions for 10 min. The samples’ turbinance were measured spectrophotometrically at 600 nm (Multiskan GO 1510, Thermo Fisher Scientific, Vantaa, Finland). Additional studies with pure CS gel base with HTZ, PO with HTZ, or pure drug were performed concomitantly. Additional controls were applied in the studies to exclude the impact of excipients on anti-inflammatory activity, namely:(a)Control 1 (B1): enzymes and hyaluronic acid were replaced with an acetate buffer (25.0 μL), and the extract was replaced with the extraction mixture (10.0 μL);(b)Control 2 (B2): the enzyme solution was exchanged with an acetate buffer (25.0 μL), and the extract was replaced with the extraction mixture (10.0 μL);(c)Control 3 (B3): the extract was replaced with the extraction mixture (10.0 μL);(d)Control 4 (B4): the hyaluronic acid solution was replaced with an acetate buffer (25.0 μL); and(e)Control 5 (B5): the enzyme solution was replaced with an acetate buffer (25.0 μL).

The percentage of hyaluronidase inhibition (I%) was calculated as follows:(1)$$I\%=\frac{\left(A_{S}-A_{B4}\right)-\left(A_{B3}-A_{B1}\right)}{\left(A_{B5}-A_{B4}\right)-(A_{B3}-A_{B1})}*100\;\%$$
where *A_S_*—absorbance of the sample, *A_B_*_1_—absorbance of blank 1, *A_B_*_3_—absorbance of blank 3, *A_B_*_4_—absorbance of blank 4, and *A_B_*_5_—absorbance of blank 5.

### 3.10. Agarose Overlay Assay 

In vitro cytotoxicity studies were performed by agarose overlay assay in mouse fibroblasts in accordance with ISO 10993-5:2009 [45]. The mouse fibroblasts (ATCC CCL, NCTC clone L929) were cultured in a minimum essential medium (MEM) and supplemented with penicillin (100 U/mL), streptomycin (100 mg/mL), 2 mmol/L L-glutamine, and 5% fetal bovine serum (all from Gibco BRL, Paisley, UK). Incubation was carried out at 37 ± 1 °C and 95% relative humidity in an air atmosphere containing 5% CO_2_. The cells were next transferred to six-well plates (1 × 10^4^ per well) and subsequently cultured for 24 h to form a confluent monolayer. Then, the cells were gently covered with 2 mL of 1% agarose in MEM. Each emulgel sample (E-1, E-2, E-3) or the corresponding drug-free formulations (P-1, P-2, P-3) were applied (with volume 10 μL) on a 6 mm disc, placed in the center of a well, and incubated for 24 h at 37 ± 1 °C. After that time, 2 mL of 0.5 mg/mL MTT solution was added to each well and incubated for 2 h at 37 ± 1 °C. The non-stained lysis zone was then examined according to the grading presented in Table 6. Phosphate buffer saline was applied as the negative control (NC), and 3% sodium dodecyl sulfate was used as the positive control (PC—3% SDS). The studies were performed in quadruplicate. The morphological analyses of L929 cells after incubation with emulgel samples were carried out with an optical microscope MW 100 equipped with a MI6 camera (Opta-Tech, Warsaw, Poland), and images were taken using Capture software version 2.3.

### 3.11. Statistical Analysis

The results were analysed using the StatSoft Statistica 13.0 software (StatSoft, Kraków, Poland). The results were presented as arithmetic means ± standard deviations (S.D.). The normality of the variable distribution was checked with the Shapiro–Wilk test, and the homogeneity of variance was checked using the Levene’s test. Data from the dissolution, and retention studies were investigated using one-way analysis of variance (ANOVA) with a post-hoc Tukey’s test. Measurements were considered significant at *p* < 0.05.

## 4. Conclusions

The designed emulgels comprised of a chitosan gel base and pomegranate seed oil appear to be promising carriers for the effective treatment of inflammatory topical diseases. All formulations had superior HTZ release as compared to conventional topical preparation and showed negligible reduction in mouse fibroblasts’ L929 cell viability, confirming their non-irritancy and compatibility with skin cells. The obtained findings point to the complex nature of HTZ release and permeability from designed emulgels, which depend on the chitosan to pomegranate seed oil weight ratio. While the higher ratio of chitosan was found to enhance the drug dissolution and penetration rate, the pomegranate seed oil was responsible for a greater degree of the emulgels’ anti-inflammatory activity. Overall, the formulation E-2 with a CS/PO weight ratio of 6:4 showed adjunctive biological activity by inhibiting the hyaluronidase enzyme and facilitating the drug release rate, and accumulation in skin was found to be most valuable among those tested. Further in vivo studies should examine the therapeutic potentials and safety profiles of selected HTZ-loaded emulgels toward inflammatory topical diseases, including contact and atopic dermatitis.

## Figures and Tables

**Figure 1 ijms-25-03765-f001:**
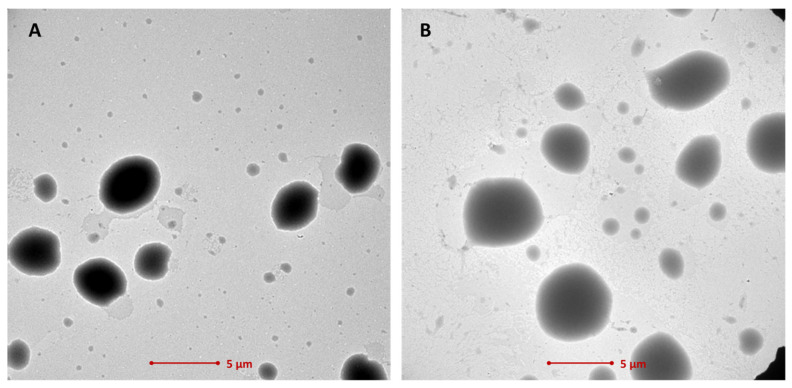
Representative TEM micrographs of (**A**) emulgel E-1 with CS/PO in the weight ratio 8:2 and (**B**) emulgel E-2 with CS/PO in the weight ratio 6:4; the scale bar is 5 μm.

**Figure 2 ijms-25-03765-f002:**
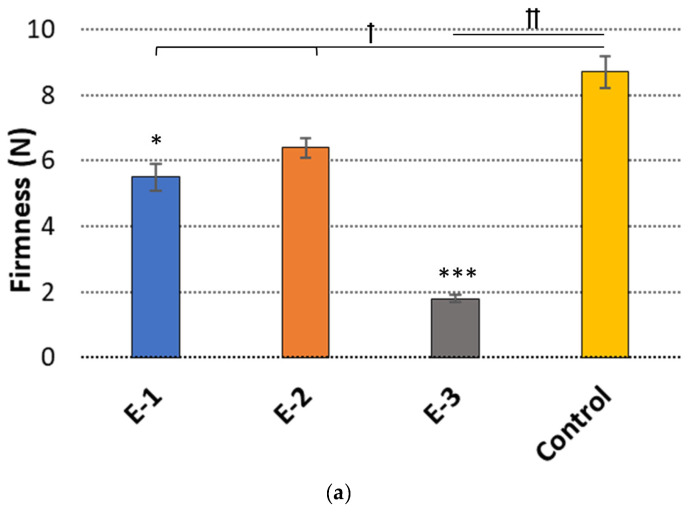
Textural behavior: (**a**) hardness, (**b**) cohesiveness, and (**c**) consistency of the emulgels with HTZ differing in the CS/P weight ratios as compared to the control (reference topical cream) (*n* = 3; mean ± S.D.). *, **, and *** represent differences between the HTZ-loaded formulations with *p* ≤ 0.05, *p* ≤ 0.01, and *p* ≤ 0.001, respectively; † and †† symbolize differences with *p* ≤ 0.05 and *p* ≤ 0.01, respectively, as compared to the control.

**Figure 3 ijms-25-03765-f003:**
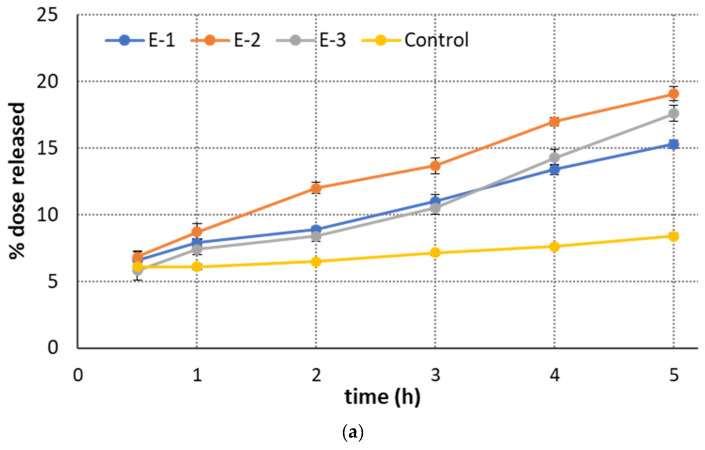
In vitro HTZ dissolution profiles: (**a**) initial stage of the studies and (**b**) emulgels differing in chitosan/pomegranate oil weight ratio compared to the reference commercial cream with HTZ acetate (Control); mean ± S.D.; *n* = 3.

**Figure 4 ijms-25-03765-f004:**
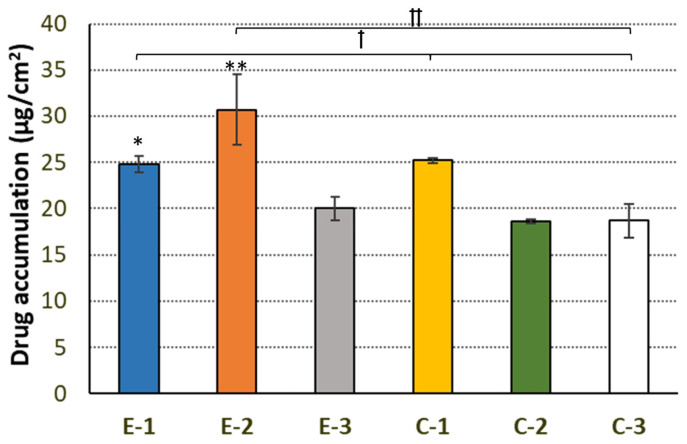
In vitro retention (expressed as the amount of drug per membrane surface area) of HTZ embedded within the emulgel formulations E-1–E-3 in StratM membrane measured after 24 h. Controls: CS gel base with HTZ (C-1), PO with HTZ (C-2), and pure HTZ (C-3). Mean ± S.D; *n* = 4. * and ** represent differences between the HTZ-loaded formulations with *p* ≤ 0.05 and *p* ≤ 0.01, respectively; † and †† symbolize differences with *p* ≤ 0.05 and *p* ≤ 0.01, respectively, as compared to the control.

**Table 1 ijms-25-03765-t001:** Composition of emulgels with hydrocortisone (HTZ).

Component (%)	Formulation			
	Emulgel o/w		Emulgel w/o	
	E-1	E-2	E-3	E-4
HTZ	1.0	1.0	1.0	1.0
Glycerolum	1.0	1.0	1.0	1.0
Lecithin	5.0	5.0	5.0	5.0
Cetostearic acid	-	2.0	2.0	2.0
Pomegranate oil	20.0	40.0	80.0	60.0
CS 3% (*w*/*w*) in 1% lactic acid	80.0	60.0	20.0	40.0

**Table 2 ijms-25-03765-t002:** Physicochemical parameters of HTZ-loaded emulgels that differ in the weight ratio of chitosan hydrophilic gel base (CS) to pomegranate seed oil (PO).

Parameter	Formulation			
	Emulgel o/w		Emulgel w/o	
	E-1 CS/PO 8:2	E-2 CS/PO 6:4	E-3 CS/PO 2:8	E-4 CS/PO 4:6
pH *	5.7 ± 0.1	5.7 ± 0.1	5.9 ± 0.1	5.8 ± 0.1
Drug content uniformity (%) *	105.6 ± 4.2	98.6 ± 1.8	93.1 ± 2.9	92.7 ± 7.1
Soluble fraction of HTZ (%) *	25.1 ± 0.2	25.7 ± 0.1	27.2 ± 0.3	26.6 ± 0.2
Physical stability upon centrifugation	uniform	uniform	phase separation	phase separation

* mean ± S.D.; *n* = 3.

**Table 3 ijms-25-03765-t003:** Values of HTZ’s apparent permeability coefficients from CS/PO emulgels that differ in CS to PO weight ratios (E-1–E-3) and controls (Control 1—pure CS gel base with HTZ; Control 2—pure PO with HTZ; Control 3—pure drug).

Time	E-1	E-2	E-3	Control 1	Control 2	Control 3
	*P_app_* × 10^−6^ cm/s					
4 h	0.30 ± 0.01 ^b^	0.30 ± 0.02 ^b^	0.59 ± 0.15 ^a^	0.10 ± 0.03 ^b^	0.72 ± 0.22 ^a^	0.23 ± 0.16 ^b^
24 h	3.04 ± 0.85 ^a^	2.22 ± 0.84 ^a^	1.86 ± 0.62 ^a^	0.67 ± 0.39 ^b^	0.83 ± 0.08 ^b^	0.35 ± 0.24 ^b^

Mean values within a row with the same letter are not significantly different at *p* < 0.05 using Duncan’s test. The first letter of the alphabet indicates the highest value (a) followed by statistically significant decreasing values (b).

**Table 4 ijms-25-03765-t004:** In vitro anti-inflammatory activity of CS/PO emulgel formulations with HTZ (E-1–E-3) as compared to the controls: CS gel base with HTZ (Control 1); PO with HTZ (Control 2); and pure HTZ (Control 3). Mean ± S.D; *n* = 3.

E-1	E-2	E-3	Control 1	Control 2	Control 3
IC_50_ [mg emulgel/mL]					IC_50_ [mg/mL]
76.03 ± 2.47 ^c^	19.46 ± 1.41 ^a^	18.69 ± 0.15 ^a^	16.91 ± 2.19 ^a^	63.47 ± 1.56 ^b^	>1

Mean values within a row with the same letter are not significantly different at *p* < 0.05 using Duncan’s test; the first letter of the alphabet is for the highest activity (a), the next is for statistically significant decreasing activity (b, c).

**Table 5 ijms-25-03765-t005:** Cytotoxic effects of the designed emulgels E-1, E-2, and E4, differing in CS/PO weight ratios, evaluated by agarose overlay assay as compared to drug-free formulations (P-1, P-2, P-3) and controls: PC–3% SDS and NC–PBS. Mean ± S.D; *n* = 3.

Sample	Diameter of Lysis Area	Area of Lysis Zone	Morphological Assessment	Grade
PC	21.34 mm	3.87 cm^2^	All cells were lysed within the area	4
NC	None	None	Cells without any morphological changes	0
E-1	None	None	Cells under the filter disc with a sign of slight degeneration	1
E-2	None	None	Cells without any morphological changes	0
E-3	None	None	Cells without any morphological changes	0
P-1	None	None	Cells without any morphological changes	0
P-2	None	None	Cells without any morphological changes	0
P-3	None	None	Cells without any morphological changes	0

**Table 6 ijms-25-03765-t006:** Qualitative morphological grading of cytotoxicity in accordance with ISO 10993-5:2009.

Grade	Reactivity	Condition of a Culture and Lysis Zone
0	None	No cell lysis zone
1	Slight	Slight growth inhibition, cells partially degenerated under the filter disc
2	Mild	Higher growth inhibition, no more than 50%, lysis zone restricted to the place under the filter disc
3	Moderate	More than 50% of growth inhibition, radius of lysis zone <1 cm around the filter disc
4	Severe	Nearly complete or complete destruction of cells, radius of lysis zone >1 cm around the filter disc

## Data Availability

Data are contained within the article and Appendix A.

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
