# Peer review of "Chitosan/Pomegranate Seed Oil Emulgel Composition as a New Strategy for Dermal Delivery of Hydrocortisone"

_ijms, 2024, doi:10.3390/ijms25073765_

Round 1

Reviewer 1 Report

Comments and Suggestions for Authors

The paper titled "Chitosan/Pomegranate Seed Oil Emulgel Composition As a New Strategy for Dermal Delivery of Hydrocortisone" introduces a novel approach in the field of dermal delivery systems by focusing on the formulation of an emulgel using chitosan and pomegranate seed oil as key components. This composition stands out as it combines the unique properties of chitosan, a biocompatible and biodegradable polymer, with pomegranate seed oil, known for its antioxidant and anti-inflammatory properties. By utilizing these ingredients, the paper addresses a specific gap in the field concerning the effective delivery of hydrocortisone, a widely used corticosteroid, through the skin. This approach holds promise for enhancing the therapeutic efficacy of hydrocortisone while minimizing potential side effects associated with conventional delivery methods. Furthermore, it is important to highlight the significance of natural ingredients in dermatological formulations, paving the way for safer and more efficient treatments in dermatology.

The methodology employed in formulating the emulgels demonstrates a meticulous and well-grounded approach, ensuring reproducibility and reliability of the obtained results. The controls conducted, both in terms of emulgel characterization and evaluation of its efficacy in dermal hydrocortisone delivery, reflect a rigorous and comprehensive scientific approach, providing a solid theoretical and contextual support for the study. Furthermore, the references cited in the paper are relevant and up-to-date.

Due to the above comments, I consider the article ready for publication in International Journal of Molecular Sciences after making the following minor modifications:

1.       Lines 112-117. The authors explain why E—4 formulation is excluded from further examinations, however, E-3 formulation, while having limited physical stability as E-4, is still studied in some cases, but not in others, such as TEM characterization. The authors should further clarify these decisions.

2.       Some images of the experiments to support the data in the graphs and tables are missing. The authors should include, for example, the images corresponding to the results presented in Table 4, even if only in the supplementary information.

Comments on the Quality of English Language

Minor editing of English language required

Author Response

Comment 1. Lines 112-117. The authors explain why E—4 formulation is excluded from further examinations, however, E-3 formulation, while having limited physical stability as E-4, is still studied in some cases, but not in others, such as TEM characterization. The authors should further clarify these decisions.

Reply. We appreciate comment raised by Reviewer. Upon characterization studies we noticed that all formulations remained homogenous with no signs of separation up to 30 days of storage except for formulation E-4 which spontaneously separated within 24h after preparation. Further studies with centrifugation step confirmed lack of stability of E-4 (and therefore we excluded it from further studies) but also showed phase separation in formulation E-3 (w/o). Despite sensitivity to mechanical stress we decided to continue our research work with emulgel E-3 as it helped us better distinguish the correlations between chitosan and pomegranate seed oil and gave us better insight into relatioships between CS/PO ratio and emulgels properties. We would like to confirm that formulation E-3 was included in all experimental studies except for TEM examination where dilution step with aqueous solution of phosphotungstic acid (at the sage of sample preparation for microscopic analysis) caused phase separation of emulgel w/o E-3 that disabled proper morphological examination. According to Reviewer remark, we rephrased statement to clearly justify our choice (page 3, line 119).

Comment 2. Some images of the experiments to support the data in the graphs and tables are missing. The authors should include, for example, the images corresponding to the results presented in Table 4, even if only in the supplementary information.

Reply. Taking into consideration the Reviewer remark, images of cells upon incubation with emulgels E-1 – E-3 and controls  were included in Supplementary material to support data from Results &discussion section. Methodology section was updated with details from microscopic analysis (page 14, line 511).

The authors are pleased that the Reviewer feels that the manuscript is valuable and draws interesting findings.

Reviewer 2 Report

Comments and Suggestions for Authors

1. Why did the authors study different formulations of CS/PO (8:2,6:4,2”8, and 4:6)? Why did the authors not fix the same formulation compared to the referent formula?

2.  Figure 3b does not scale on the x-axis, making it difficult to pinpoint the correct positions.

3.   There are many chemical substancs mentioned in the Materials and Methods section, such as CTAB, phophotungstic acid hydrate, potassium dihydrogen phosphate, sodium laurylsulphate (SDS), and water. However, they are not shown in Table 5.

4.   What important substance and structure does pomegranate seed oil contain?

Author Response

Comment 1. Why did the authors study different formulations of CS/PO (8:2,6:4,2”8, and 4:6)? Why did the authors not fix the same formulation compared to the referent formula?

Reply. Thank you for the comment. Presented data was a preliminary assessment of topical emulgels which aimed at finding the most suitable chitosan to pomegranate seed oil ratio in terms of enhanced hydrocortisone delivery. At development stage we could not decide which formula could be optimal therefore we decided to examine all designed preparations to assess the finest ratio of key ingredients. Overall, collected data from pharmaceutical and biological studies helped us discriminate the most desirable formulation for further research work. According to Reviewer remark, we rephrased and clarified the scope of work (page 2, line 91).

Comment 2.Figure 3b does not scale on the x-axis, making it difficult to pinpoint the correct positions.

Reply. Fig. 3b was improved according to Reviewer suggestion. We apologize for this oversight.

Comment 3. There are many chemical substances mentioned in the Materials and Methods section, such as CTAB, phophotungstic acid hydrate, potassium dihydrogen phosphate, sodium laurylsulphate (SDS), and water. However, they are not shown in Table 5.

Reply. Materials stated in Section 3.1. referred to all reagents, buffer ingredients and chemicals used in the studies. In turn, Table 5 (in revised manuscript, Tab. 1) presented solely the emulgel composition.

Comment 4.What important substance and structure does pomegranate seed oil contain?

Reply. We appreciate the comment raised by reviewer. Pomegranate seed oil contains a variety of active ingredients, including phenolic acids, flavonoids, phytosterols, or gamma-tocopherol  and probably the most important and unique one - punicic acid (representing conjugated fatty acids) considered as the major compound responsible for its biological activity. Accoridng to Reviewer remark, proper information was included in Introduction section (page 2, line 66).

We appreciate all the comments raised by the Reviewer which helped to improve the quality of the manuscript.

Reviewer 3 Report

Comments and Suggestions for Authors

The research article entitled "Chitosan/Pomegranate Seed Oil Emulgel Composition As a New Strategy for Dermal Delivery of Hydrocortisone" introduces a potentially effective method for delivering hydrocortisone through topical application in the treatment of inflammatory skin disorders. The researchers have created emulgels by mixing chitosan (CS) and pomegranate seed oil (PO). These emulgels not only help release hydrocortisone, a model anti-inflammatory medication, but also show additional pharmacological activity. The results underscore the significance of the CS/PO ratio in regulating drug absorption and biological activity. Hydrocortisone absorption is predominantly influenced by CS, while the anti-inflammatory action is enhanced by PO through the inhibition of hyaluronidase activity. The utilization of a CS/PO ratio of 6:4 in the formulation demonstrates notable efficacy, presenting a promising approach for the management of inflammatory skin conditions. In summary, this research highlights the ability of multifunctional delivery systems to enhance the effectiveness of drugs and reduce skin irritation, making a noteworthy addition to the field of dermatological treatments. All of these findings seem valuable to be an acceptable research work and suitable for the publication in IJMS after justifying the following points:

·        The similarity rate are high and should be rephrased in the method section  

·        In the abstract “The complex nature of drug accumulation and release rate from tested emulgels was noticed and clear effect of CS/PO on the emulgel biological activity was observed” it is recommended to improve these results with values for your combination and the positive and/or negative controls.

·        Page 1, line 44-45 “Hydrocortisone (HTZ) is a low potent steroid agent, sparingly soluble in aqueous media (250 μg/ml) with log 1.61” I think you mean Log S ?? you have to edit it.

·        Page 2 line 67-69 it is recommended to add an example for these strategy with recent work like “Synthesis of novel isoxazole–carboxamide derivatives as promising agents for melanoma and targeted nano‑emulgel conjugate for improved cellular permeability (https://doi.org/10.1186/s13065-022-00839-5), where the authors improved the cell permeability by using similar strategy.

·        You can improve the introduction with recent works regarding the emulgel for some plant oils by using the following recent work too “Coatings 2023, 13, 1441” and “https://doi.org/10.1186/s12906-021-03324-z” all of these works can be added in the introduction or discussion sections

·        All of your results were compered with Control which is an “conventional formulation cream” so it is recommended to write “positive control” because this formulation has the active ingredient too

·        In figure 3a you have to start your concentration points from 0 value, like figure 3b

·        Line 291 correct the ref. No 36

·        In Table 3 add the explanation as footnote for a, b and c letters

·        All Tables have a result should be moved to the results section not in method section

·        The conclusion could be improved by adding the next step future works

Best Regards 

Author Response

Comment 1. The similarity rate are high and should be rephrased in the method section

Reply. We appreciate the comment raised by Reviewer. Methodology section was modified and shortened to avoid unnecessary similarity. 

Comment 2. In the abstract “The complex nature of drug accumulation and release rate from tested emulgels was noticed and clear effect of CS/PO on the emulgel biological activity was observed” it is recommended to improve these results with values for your combination and the positive and/or negative controls

Reply. According to reviewer remark, we rephrased the abstract section as follows (page 1, line 24):

“(…) All formulations favored hydrocortisone release with up to two-fold increase in drug dissolution rate within first 5h when compared to conventional topical preparation. The clear effect of CS/PO on the emulgel biological performance was observed and CS was found prerequisite for modulation hydrocortisone absorption and accumulation. In turn, greater amount of PO played the predominant role in inhibition of hyaluronidase activity and enhanced anti-inflammatory effect of preparation E-3.”

Comment 3.Page 1, line 44-45 “Hydrocortisone (HTZ) is a low potent steroid agent, sparingly soluble in aqueous media (250 μg/ml) with log 1.61” I think you mean Log S ?? you have to edit it.

Reply. Thank you for this remark. Actually, it is Log P value. Introduction section was updated (page 1, line 45).

Comment 4. You can improve the introduction with recent works regarding the emulgel for some plant oils by using the following recent work too “Coatings 2023, 13, 1441” and “https://doi.org/10.1186/s12906-021-03324-z” all of these works can be added in the introduction or discussion sections

Reply. Proper reference was added to introduction section according to Reviewer remark (ref 8).

Eid, A. M.; Hawash, M.; Abualhasan, M.; Naser, S.; Dwaikat, M.; Mansour, M. Exploring the Potent Anticancer, Antimicrobial, and Anti-Inflammatory Effects of Capparis Spinosa Oil Nanoemulgel. Coatings 2023, 13 (8), 1441. https://doi.org/10.3390/coatings13081441.

Comment 5. Page 2 line 67-69 it is recommended to add an example for these strategy with recent work like “Synthesis of novel isoxazole–carboxamide derivatives as promising agents for melanoma and targeted nano‑emulgel conjugate for improved cellular permeability (https://doi.org/10.1186/s13065-022-00839-5), where the authors improved the cell permeability by using similar strategy.

Reply. Proper reference was added to introduction section according to Reviewer remark (ref 10).

Hawash, M.; Jaradat, N.; Eid, A. M.; Abubaker, A.; Mufleh, O.; Al-Hroub, Q.; Sobuh, S. Synthesis of Novel Isoxazole–Carboxamide Derivatives as Promising Agents for Melanoma and Targeted Nano-Emulgel Conjugate for Improved Cellular Permeability. BMC Chemistry 2022, 16 (1), 47. https://doi.org/10.1186/s13065-022-00839-5.

Comment 6.  All of your results were compared with Control which is an “conventional formulation cream” so it is recommended to write “positive control” because this formulation has the active ingredient too.

Reply. We agree with that reference HTZ cream may be considered positive as it contains drug substance. However we did not use ‘positive control’ for this preparation as this term was used in the cytotoxicity studies. According to ISO requirements, positive control commonly refers to cytotoxic substance with proved irritation properties. In order to not mislead the readers we did not decide to apply commercial product as ‘positive’ in textural and dissolution studies.

Comment 7.  In figure 3a you have to start your concentration points from 0 value, like figure 3b

Reply. We appreciate the comment. We would like to confirm that the initial time point in both fig. 3 a and 3b is the same (and amounts to 30min). Keeping in mind that Fig. 3b might be blurred due to high density of data points at the beginning of test we decided to detach release profile within first 5h as a separate fig.

Comment 8. Line 291 correct the ref. No 36

Reply. Style of ref. was corrected. Apology for this oversight.

Comment 9.   In Table 3 add the explanation as footnote for a, b and c letters

Reply. Footnote of Tab. 3 (in revised version, Tab.4) was updated (page 10, line 312). Apology for this oversight.

Comment 10.   All Tables have a result should be moved to the results section not in method section

Reply. According to Reviewer remark, Table 5 (in revised manuscript Tab.1) was moved to Results and Discussion Section (page 3, line 105). We decided to keep Table 6 in Methodology as it displayed a brief summary of data interpretation not results.

Comment 11.  The conclusion could be improved by adding the next step future works 

Reply. Thank you for this comment. Additional remark was included in Conclusion section (page 15, line 537).

We appreciate all the comments raised by the Reviewer which helped to improve the quality of the manuscript.

Round 2

Reviewer 3 Report

Comments and Suggestions for Authors

The authors were improved the manuscript according to all suggested comments, no further comments 

Best wishes